# Classification and Functional Analysis between Cancer and Normal Tissues Using Explainable Pathway Deep Learning through RNA-Sequencing Gene Expression

**DOI:** 10.3390/ijms222111531

**Published:** 2021-10-26

**Authors:** Sangick Park, Eunchong Huang, Taejin Ahn

**Affiliations:** 1Department of Advanced Convergence, Handong Global University, Pohang-si 37554, Gyeongbuk, Korea; 21932001@handong.edu (S.P.); hec1324@gmail.com (E.H.); 2Department of Life Science, Handong Global University, Pohang-si 37554, Gyeongbuk, Korea

**Keywords:** deep learning, neural networks, biological function, pathway, cancer gene expression

## Abstract

Deep learning has proven advantageous in solving cancer diagnostic or classification problems. However, it cannot explain the rationale behind human decisions. Biological pathway databases provide well-studied relationships between genes and their pathways. As pathways comprise knowledge frameworks widely used by human researchers, representing gene-to-pathway relationships in deep learning structures may aid in their comprehension. Here, we propose a deep neural network (PathDeep), which implements gene-to-pathway relationships in its structure. We also provide an application framework measuring the contribution of pathways and genes in deep neural networks in a classification problem. We applied PathDeep to classify cancer and normal tissues based on the publicly available, large gene expression dataset. PathDeep showed higher accuracy than fully connected neural networks in distinguishing cancer from normal tissues (accuracy = 0.994) in 32 tissue samples. We identified 42 pathways related to 32 cancer tissues and 57 associated genes contributing highly to the biological functions of cancer. The most significant pathway was G-protein-coupled receptor signaling, and the most enriched function was the G1/S transition of the mitotic cell cycle, suggesting that these biological functions were the most common cancer characteristics in the 32 tissues.

## 1. Introduction

Cancer is one of the most aggressive diseases worldwide, accounting for nearly nine million deaths universally. One of the defining features of cancer is the rapid duplication of abnormal cells beyond their usual boundaries, then invading adjoining parts of the body. Various efforts have been conducted in terms of deep learning for the classification and interpretation of cancer especially by using medical image as major data source [1,2,3]. As cancer is linked to its host’s genomic information, the method of analyzing pathways related to cancer-specific genes has been the traditional research direction. With the recent development of high-throughput RNA-sequencing (RNA-seq), research combining cancer gene expression and deep learning technology has gained increasing attention [4,5].

Although deep neural network (DNN)-based research helps to resolve goal-oriented problems, its use in biological interpretation is limited [6]. To provide comprehensive biological interpretation, here have been several attempts to apply pathway data structures in the deep learning approach. DeepCC uses gene set enrichment scores as a pathway-based normalized input value to distinguish four subtypes of colorectal cancer [7]. MSigDB 6.0 (17,779 gene sets) was used to provide an enrichment score via gene set enrichment analysis (GSEA) [8]. Colorectal cancer (*n* = 626) RNA-seq data from The Cancer Genome Atlas (TCGA) were used as training datasets [9]. Microarray gene expression data from the Gene Expression Omnibus (GEO; colorectal cancer, 13 datasets, *n* = 2952) were used for validation [10]. DeepCC exhibited higher efficacy with GEO than other machine learning methods, proving that using GSEA as normalized input for the DNN may help reduce the limitation of data usability, likely due to the platform difference between RNA-seq and microarray data [11].

Another study using deep learning and pathway information used the Kyoto Encyclopedia of Genes and Genomes (KEGG) BRITE hierarchy to image gene expression. These images were the input for a convolutional network [12]. This approach predicted the progression-free interval (PFI) of lung cancer [13]. This approach used 10,535 samples and 7509 features of 33 TCGA pan-cancer types and pre-trained a convolutional neural network using gene-expression images of non-lung cancer. The PFI prediction performance of lung cancer AUC 0.7326 was obtained by fine-tuning the pre-trained convolutional neural network with lung cancer gene expression images [14]. Other methods, including logistic regression, support vector machine (SVM), and random forest (RF), have not achieved such performances [15,16].

PathDNN is a DNN that comprises gene-to-pathway relationship data from 323 KEGG pathways in the first hidden and input layers [17]. PathDNN predicts the drug sensitivity of cancer cells when a drug and cell line is given, using gene expression data. It uses gene expression (17,737 genes, 970 cancer cell lines) data from the Genomics of Drug Sensitivity in Cancer database and drug targets (250 drugs, 1100 target proteins) from the “search tool for interactions of chemicals” [18,19]. The Pearson correlation coefficient of drug sensitivity predicted by PathDNN was 0.8, which is higher than other machine learning methods.

The Gene–pathway–disease (the GPD model) model connects input genes to the first layer, comprising of 1708 reactome pathways, to the output layer, representing the disease phenotype as a node; thus, the disease phenotype is predicted [20]. Using 4788 microarray samples, 9247 genes, and 83 types of diseases, the the GPD model model predicts association scores between diseases and genes or pathways by quantifying the changes in disease scores induced by dysregulations in the expression of genes or activation of pathways (sensitivity analysis). Sensitivity analyses evaluating drugs and disease-associated gene expression and pathway activation are performed to identify diseases related to genes or pathways. In the GPD model, the AUC for disease–gene association prediction has been reported as 0.59, whereas the AUC for disease–disease association prediction is 0.62 for both genes and pathways; thus, the AUCs generated by the GPD model are better than those produced by other machine learning techniques.

PASNet connects genes to a node that represents a biological pathway, which is further connected to fully connected hidden layers [21]. This method involves a sparse coding concept that discards unnecessary connections in hidden layers. During training, it calculates the weights of nodes and updates only nodes with a strong influence as learning continues. PASNet provides a certain level of interpretability because it can select nodes with more substantial influence in each layer. This system has been applied to predict survival in glioblastoma multiforme (GBM) in a binary manner, with long and short survival periods. The AUC obtained by PASNet is 0.6622, which is superior to that obtained by other machine learning techniques.

Previous studies including those utilizing PathDNN, DeepCC, the GPD model, and PASNet and the study of López-García et al., made use of pathway knowledge in the DNN structure. Differences between these approaches and our proposed method are examined below. PathDNN uses the pathway structure for DNN, but it learns the relationship between the drug and drug target as well as the gene expression of the drug-administered cell. PathDNN applies only to drug efficacy prediction, which is different from the purpose of PathDeep. DeepCC and the method described by López-García et al. use genes-to-pathway information for deep learning training, but they do not provide both the pathway contribution index and a comprehensive explanation of their member genes. The GPD model provides the pathway index and performs perturbations to change the pathway node value when analyzing the association between the pathway and phenotype. The GPD model differs from our approach because PathDeep searches for statistically significant pathways using permutations of the phenotype label. PASNet provides a calculation method different from PathDeep for the association between the pathway and phenotype. PASNet summates all the weights from the pathway node to the output node as an index, but PathDeep uses the pathway node value itself as an index.

In this study, we propose PathDeep (https://github.com/sipark5340/PathDeep.git, accessed on 21 October 2021), which uses the pathway structure to learn the gene difference between cancer from normal tissues and provides a pathway based interpretation of it (Pathway index and pathway contribution gene index). We aimed to study the performance of PathDeep in classifying cancers from normal tissues based on gene expression data. PathDeep showed a favorable performance and provided a pathway knowledge-based interpretation.

## 2. Results

### 2.1. Selecting the Pathway Database for the PathDeep Approach

We evaluated classification performance using different pathway databases by representing its gene-to-pathway relationships in the PathDeep structure. We also assumed that the gene-to-pathway relationship is random (PathDeep random linked), connecting the gene nodes to a pathway node at random but with the same number of edges that a pathway database provides. We repeatedly generated this random network and then trained each network to reveal that the biological relationship model demonstrates better performance than random networks. Among 15 pathway databases considered, the c2 reactome showed the highest test performance (accuracy = 0.994) and the lowest *p* value when comparing null distribution, which assumes all gene-to-pathway relationships in DNN are random (*p* = 0.0519). Thus, we concluded that the c2 reactome provides more information when discriminating cancer from normal tissues than other pathway databases. We chose the c2 reactome database for further analysis (Table 1).

### 2.2. PathDeep Performance Compared with That of Other Methods

The average test accuracy of PathDeep was 0.9928, which is the best among all the experimented methods, including other machine learning methods examined in this study. PathDeep provided higher accuracy than fully connected DNNs (*p* < 2.22 × 10^−16^). It also showed higher accuracy than PathDeep random linked. The results show that the representation of PathDeep of the biological relationship between genes and pathways contributed to its training performance. PathDeep performed better than the MLNN (*p* < 2.22 × 10^−16^). The average test accuracy of other machine learning methods—SVM, Extreme Gradient Boosting Machine (XGboost), Gradient Boosting Machine (GBM), and RF—were 0.9925, 0.9921, 0.9905, and 0.9895, respectively [22,23]. The performance of PathDeep was better than SVM, which showed the best performance among all the machine learning methods (*p* < 2.22 × 10^−16^). This shows that the performance of a DNN with a biological relationship between the input gene and the pathway node is higher than that of a DNN with a fully connected layer. Furthermore, it is superior to other machine learning methods (Figure 1).

### 2.3. Sample Visualization Using Gene Expression and Pathway Index

The pathway index value of PathDeep and the pathway member gene expression of the c2 reactome molecular signature collection were visualized (Figure 2). This result shows that when the sample dimension is reduced from the gene level (5796 genes) to the pathway level (674 pathways), the images of the gene and pathway levels are similar in shape. The tissue of origin of the clustered samples is visually presented at both the gene and pathway levels. Therefore, the original gene expression value and pathway index show the same characteristics as that of the sample.

### 2.4. Functional Analysis Using Highly Contributing Pathway Genes

We obtained pathway contribution gene indexes (Appendix A). Pathway contribution gene indexes show long-tail distribution, which is presented in Figure 3. The gene with the highest pathway contribution gene index was PLA2G10 (pathway contribution gene index = 1.149). In addition, phospholipase A2 member genes were included among the top 1% contributing genes (PLA2G2E, PLA2G4D, PLA2G12A, and PLA2G2A). The gene with the second highest pathway contribution gene index was SHC1 (pathway contribution gene index = 0.904). The top 1% ranked genes according to the pathway contribution gene index also included histone protein genes (HIST1H3A, HIST1H4K, HIST1H4D, HIST1H4I, and HIST2H2AA4). In the functional analysis performed using the top 1% of genes with the high pathway contribution indexes, the most significant function was G1/S transition of the mitotic cell cycle (FDR = 1.03 × 10^−12^). The following four significant functions were related to DNA replication (DNA replication, DNA-dependent DNA replication, DNA strand elongation involved in DNA replication, and DNA strand elongation). The functions of the top 1% genes according to pathway contribution gene indexes are shown in Appendix A.

### 2.5. Classification of Cancer and Normal Tissues Using Key Genes with a High Pathway-Contributing Index

We evaluated the performance of PathDeep by reducing the number of gene expression values used for training by limiting the number of genes according to the pathway contribution gene index value (Figure 4). When all genes (*n* = 5796) were used, the average test accuracy of PathDeep was 0.9931. When 50% of genes (*n* = 2898) with high pathway contribution gene indexes were used, the average test accuracy of PathDeep was 0.9925. When only 1% of genes with the high pathway contribution gene indexes were used, the average test accuracy of PathDeep was 0.9741. Thus, even if only 1% of the genes (*n* = 57) with a high pathway contribution gene index were used, PathDeep could discriminate between cancer and normal phenotypes. The result shows only a few genes that PathDeep counts important is sufficient to discriminate cancer from normal.

### 2.6. Pathway Contribution Gene Indexes of the Top 57 Genes in PathDeep versus Those of a Randomly Selected 57 Genes in PathDeep

As shown in Figure 5, the average test accuracy of PathDeep in discriminating between cancer and normal phenotypes using the pathway contribution gene indexes of the top 57 genes as input was 0.9741, whereas that using the pathway contribution gene indexes of 57 random genes as input was 0.9484. There was a significant difference between the average test accuracy of PathDeep using the pathway contribution of the top 57 genes as input and that from using a randomly selected 57 genes as inputs (*p* = 2.22 × 10^−16^), indicating that the higher the pathway contribution gene index, the more information is available for the discrimination between cancer and normal phenotypes.

### 2.7. Identification of Essential Pathways for Cancer Prediction and Evaluation of Model Performance of the Essential Pathways

We identified pathways essential for cancer discrimination in a pan-cancer manner through label permutation experiments (Table 2). Of the 674 pathways originally considered, 42 pathways were found to majorly contribute toward distinguishing cancer from normal phenotypes (FDR q-value < 0.05). The most essential pathway was G-protein-coupled receptor (GPCR) downstream signaling (FDR q-value = 0). The second most important pathway was GPCR ligand binding (FDR q-value = 4.54 × 10^−100^). The average test accuracy from building 100 models with different hyperparameters of PathDeep with 42 pathways—which are related to the distinction of cancer and normal phenotypes—for the training and test of five sets was 0.9914. The average test accuracy from 100 trials of PathDeep with all 674 pathways was 0.9935; this indicates that only 42 pathways related to cancer prediction were found to be suitable for discriminating between cancer and normal tissues.

## 3. Discussion

Uncovering cancer-related biological functions could offer a multifaceted solution to researchers for treating this disease. In this study, the PathDeep model was developed for pan-cancer analysis of biological functions that have statistically significant effects on the prediction of cancer. PathDeep demonstrates why the biological knowledge of the c2 reactome is the most appropriate biological knowledge for the discrimination between cancer and normal tissues among the 15 pathway databases considered. PathDeep has a superior performance in predicting cancer and normal tissues compared to other machine learning techniques, and it also shows a higher performance of DNNs in discriminating cancer and normal tissues when a biological knowledge structure is used than a conventional machine learning approach. PathDeep identified 42 pathways related to discriminating cancer and normal tissues among 674 pathways in the c2 reactome database using representative values representing pathways. The first and second most significant pathways were REACTOME GPCR downstream signaling (FDR q-value = 0) and REACTOME GPCR ligand binding (FDR q-value = 4.54 × 10^−100^), indicating that GPCRs and their downstream signaling affect cancer growth and development [24]. Moreover, GPCRs affect numerous aspects of cancer biology, such as vascular remolding, invasion, and migration [25]. Pathways discovered using PathDeep have been identified as cancer-associated pathways in previous studies. This study demonstrates that PathDeep could analyze biological functions.

PathDeep can indicate the contribution of a gene in the pathway layer, which is an indicator of the relation of genes to a biological function. The gene with the highest pathway contribution gene index was PLA2G10 (pathway contribution gene index = 1.149), a member of the A2 phospholipases that regulate fatty acid cleavage [26]. PLA2G10 is known as a novel modulator of lipid metabolism that promotes breast cancer cell growth and survival by stimulating LD formation and FA oxidation [27]. Phospholipase A2 member genes also belong to the top 1% of the pathway-contributing gene index. Phospholipase A2 enzymes regulate the release of biologically active fatty acids and lysophospholipids from membrane phospholipid pools [28]. Fatty acids are a major source of electrons for ATP production through fatty acid oxidation (FAO) and oxidative phosphorylation (OxPhos) in cancer cells [29]. The gene with the second highest pathway contribution gene index was SHC1 (pathway contribution gene index = 0.904), which plays an important role in regulating cell apoptosis and drug resistance in numerous types of cancer [30]. In addition, overexpression of SHC adaptor proteins is associated with mitogenesis, carcinogenesis, and metastasis [31]. The top 1% of genes with a high pathway contribution gene index also included genes encoding histone proteins. Histone proteins are highly basic proteins found in eukaryotic cell nuclei that pack and order the DNA into structural units called nucleosomes [32]. The most relevant biological functions of the top 1% of pathway-contributing genes were associated with the G1/S transition of the mitotic cell cycle (FDR = 1.03 × 10^−12^). The G1/S transition occurs late in G1, and the absence or improper application of this highly regulated checkpoint can lead to cellular transformation and disease states such as cancer [33]. Other relevant biological functions of the top 1% of pathway-contributing genes are related to DNA replication and acyl-chain remodeling. Membrane lipid remodeling takes the center stage in growth factor receptor-driven cancer development [34].

In conclusion, PathDeep, constructed by utilizing biological function structure-based DNNs, showed cancer and normal tissue discrimination performance and demonstrated the possibility of biological functional analysis and explanation. PathDeep implemented the diagnosis of cancer and normal tissues for four data platforms: TCGA, TARGET, GTEx, and K-562, and performed cancer-related biological function analysis. PathDeep shows better diagnosis performance in discriminating cancer and normal tissues than other methods. Applying PathDeep with the capabilities of biological functional analysis to other data platforms and problems may represent a possibility for offering biological functional advice to various studies (e.g., drug development) and clinical practice when biological interpretation is required.

## 4. Materials and Methods

### 4.1. Gene Expression Data

For this study, we used TCGA TARGET GTEx data with the Toil RNA-seq recompute fragments per kilobase of exon per million (FPKM) version provided by the UCSC Xena database [35]. The total number of samples was 19,131, of which 10,964 and 8167 were cancer and normal samples, respectively. The total number of genes was 60,498. Gene expression levels were expressed as values obtained by adding 0.001 to FPKM on the log2 scale. An overview of the number of cancers and the TCGA, TARGET, and GTEx project data are shown in Appendix A.

### 4.2. MSigDB with TCGA TARGET GTEx

For information on pathways and member genes of pathways, a collection of 15 molecular signatures provided by MSigDB v6.0 was used. The number of pathways per molecular signature collection, the number of symbols for member genes in the pathways, and the number of member genes in pathways with common gene symbols of TCGA, TARGET, and GTEx are shown in Appendix A.

### 4.3. Data Processing and Workflow

The cancer and normal samples in the training and test datasets were stratified. We randomly selected half of the cancer and normal samples for each of the following four projects: TCGA, TARGET, GTEx, and K-562. Half of the extracted samples were used in the training sample set, and the other half were used as the test sample set. The training sample set had 9565 samples, and the test sample set was comprised of 9566 samples. The same process as above was repeated 20 times to create 20 training and test sets (Train and Test, 20 Sets). We first performed sample-wise standardize of data using the mean and standard deviation of 60,498 genes from each sample to process the input data for PathDeep to learn. In addition, we selected gene symbols for pathway members and listed 15 molecular signature collections in common with the gene symbols lists of TCGA, TARGET, and GTEx, using MSigDB. The genes selected for each molecular signature collection were the input for the PathDeep model.

We compared the test accuracy of PathDeep with that of RF, SVM, GBM, and XGboost. Furthermore, we compared the test accuracy of PathDeep with the test accuracy of the multi-layer neural network and PathDeep random linked network to prove the superior performance of PathDeep. Then, we analyzed the pathway index, pathway contribution gene index, and label permutation using PathDeep. The data processing and analysis workflow is shown in Figure 6.

### 4.4. PathDeep Model Structure

We created a DNN that includes the relationship between pathways and their member genes. To create a DNN that shows the relationship between pathways and input genes, the first layer of the DNN was defined as the pathway layer. Each node of the pathway layer represented one pathway, and only the pathway member genes among the input genes were connected to each pathway layer node. For example, among the input genes, only the member genes of the epidermal growth factor receptor (EGFR) signaling pathway were connected to the pathway layer node of the EGFR signaling pathway. After defining the first layer of the DNN as a pathway layer, fully connected layers were connected to the pathway layer to complete the PathDeep model (Figure 7), which learns the relationship between the input genes and pathway layer nodes. PathDeep models were created for each of the 15 molecular signature collections.

We created a PathDeep model with random links between gene nodes to pathway nodes. This was to investigate how the randomly generated gene-to-pathway relationship affects the discrimination between cancerous and normal tissues when compared to the linked gene nodes and pathway nodes, which are linked according to their known biological relationships in the pathway database. The total number of links between the gene nodes and pathway nodes were determined by the pathway database and was the same when the random link was generated. Therefore, in random link generation, the total number of links between gene nodes and pathway nodes are conserved, but participating gene and pathway pairs are random.

### 4.5. Selecting an Appropriate Pathway Database for PathDeep

We conducted the following experiment to identify suitable biological functional relationships for PathDeep among the 15 molecular signature collections of MSigDB. In each molecular signature collection, we implemented discriminating cancer and normal tissues using PathDeep once and using PathDeep random linked 100 times. Next, we calculated the z-score and *p* value using one sample z-test of PathDeep test accuracy using the mean and standard deviation of the 100 test accuracy of PathDeep random linked.

### 4.6. Visualizing Pathway Index

We performed R programming to analyze the library of t-SNE values and to visualize pathway layer values [36]. As a hyperparameter for t-SNE, pca = FALSE and theta = 0.0 options were used.

### 4.7. Calculating Pathway Index and Pathway Contribution Gene Index

The pathway index of the sample is defined as $P_{si}$. *P* is the pathway, *i* is the pathway number and *s* represents the sample. $g_{j}$ is gene *j*, which is a member gene of pathway *i*, and $g_{sj}\left(x\right)$ represents the expression of gene *j* of sample *s*. $W_{ij}$ is the weight equivalent of the member gene *j* of the pathway *i*. $P_{si}$ is the sum of the product of gene *j* expression of sample *s* and the equivalent weight in all samples and is a summary value of the pathway layer node of sample of PathDeep. The equation for the pathway index is shown in Equation (1):(1)$$P_{si}=\underset{j}{\sum }\left(g_{sj}\left(x\right)\cdot W_{ij}\right)+bias_{i}$$


The pathway contribution gene index is represented as $G_{j}$. *G* is the gene symbol and *j* is the gene number. $g_{j}$ is gene *j*, which is a member gene of pathway *i*, and $g_{si}\left(x\right)$ represents the expression of gene *j* of sample *s*. $W_{ij}$ is the weight equivalent to gene *j* of pathway *i*, and *N* is the number of samples. $G_{j}$ is the absolute average value of the sum of the product of the gene *j* expression of sample *s* and the equivalent weight in all samples and represents the contribution that each gene has in the pathway layer of PathDeep. The equation for the pathway contribution gene index is shown in Equation (2):(2)$$G_{j}=|(\frac{1}{N})\overset{N}{\underset{s}{\sum }}\underset{i}{\sum }\left(g_{sj}\left(x\right)\cdot W_{ij}\right)|$$


### 4.8. Functional Enrichment Test of the Top 1% Pathway-Contributing Genes

We performed GeneMANIA for biological functional analysis of the top 1% of pathway-contributing genes [37].

### 4.9. Identifying Cancer-Related Pathways Using Label Permutation

We obtained the pathway indexes using PathDeep, with the highest test accuracy in the Train & Test set 5, and calculated the average difference between the pathway indexes of cancer and normal tissues. Then, permutation of the original labels of the cancer and normal tissues in the Train & Test 5 set was performed 1000 times to obtain the permutation labels. For each permutation label, the pathway index was obtained by discriminating between cancer and normal tissues using PathDeep. A total of 1000 PathDeep learning permutation labels were conducted and the average and standard deviation of the average difference of the pathway index between cancer and normal samples were calculated. Then, we performed z-tests for each pathway index of the average difference of the original label using the average and standard deviation obtained from the permutation label (Figure 8).

## 5. Patent

Taejin Ahn, Sangick Park, and Taesung Park. Methods for diagnosis and therapeutic decision using artificial neural network trained with functional gene module and apparatus. KR Patent 10-2020-0120165 filled 18 September 2020, was based on the contents of this paper.

## Figures and Tables

**Figure 1 ijms-22-11531-f001:**
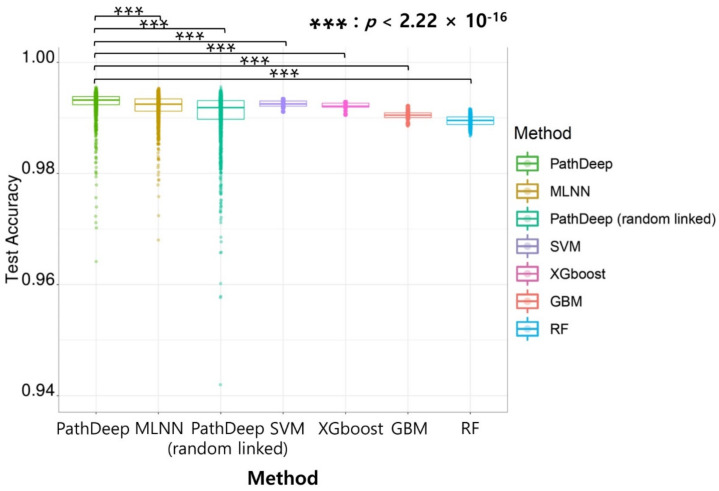
Performance of PathDeep is better than other machine learning methods.

**Figure 2 ijms-22-11531-f002:**
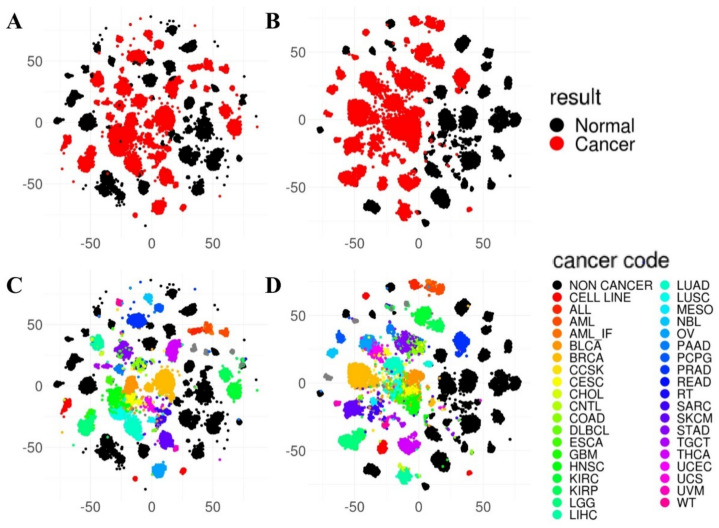
t-SNE analysis of pathway index and pathway member gene expression. (**A**) Pathway member gene expression represents both cancer and normal tissue-related information. (**B**) The pathway index also includes cancer and normal tissue-related information (e.g., pathway member gene expression data). (**C**) Pathway member gene expression contains information on cancer types. (**D**) The pathway index also contains information on cancer types, such as pathway member gene expression.

**Figure 3 ijms-22-11531-f003:**
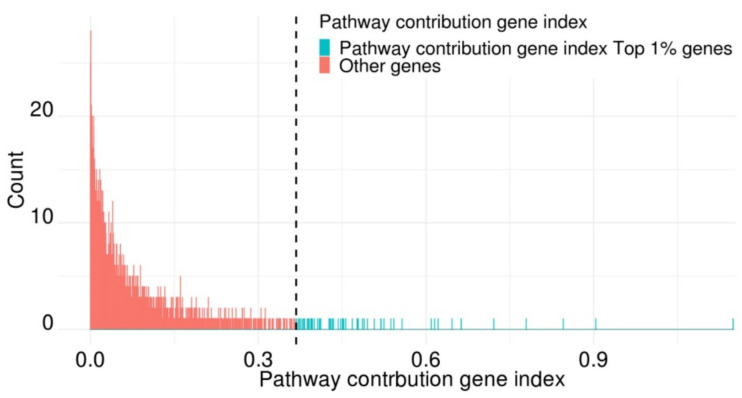
Pathway contribution gene index histogram.

**Figure 4 ijms-22-11531-f004:**
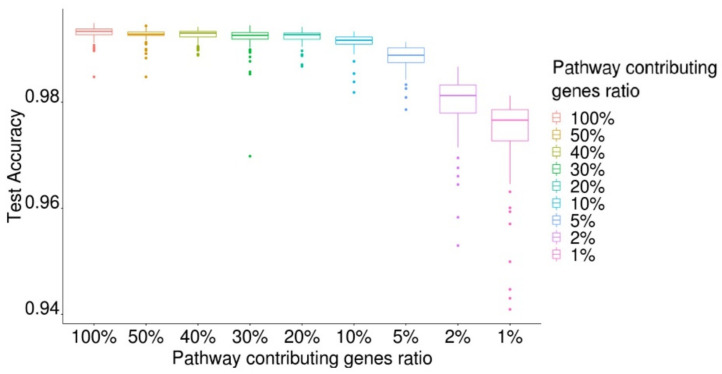
PathDeep performance using pathway-contributing gene ratio.

**Figure 5 ijms-22-11531-f005:**
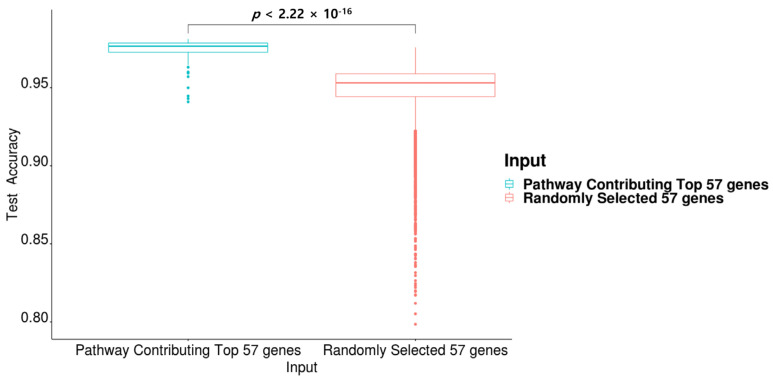
Performance comparison between pathway contribution of the top 57 genes and a randomly selected 57 genes.

**Figure 6 ijms-22-11531-f006:**
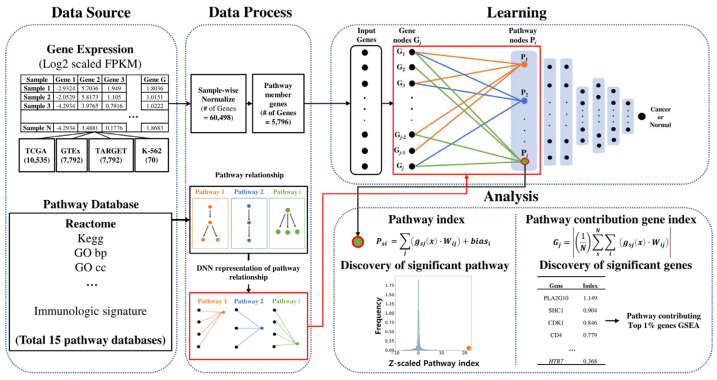
Data processing and analysis flowchart.

**Figure 7 ijms-22-11531-f007:**
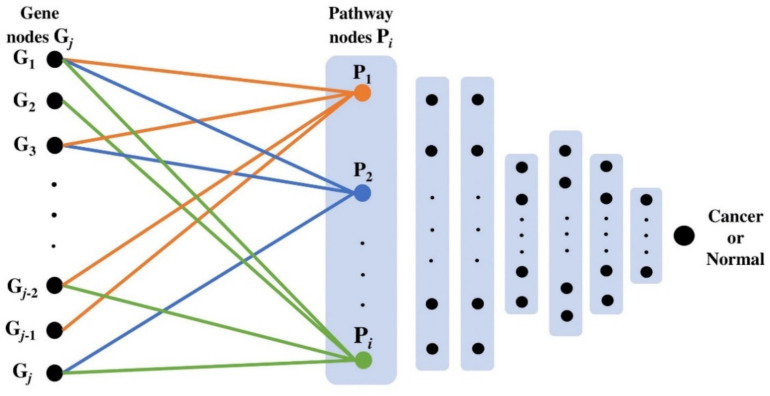
PathDeep model structure.

**Figure 8 ijms-22-11531-f008:**
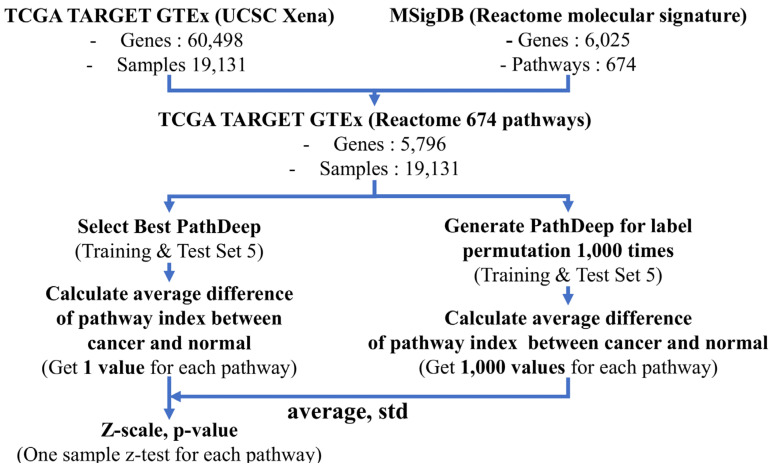
Workflow of the PathDeep label permutation study for identifying cancer-related pathways.

**Table 1 ijms-22-11531-t001:** One sample *z*-test of PathDeep performance for molecular signature collection.

Pathway Database Sources	Accuracy	*p* Value
c2 reactome	0.994	5.19 × 10^−2^
c6 oncogenic signatures	0.993	1.12 × 10^−1^
c5 GO cc	0.993	1.57 × 10^−1^
c2 kegg	0.992	2.40 × 10^−1^
c3 tft	0.991	2.49 × 10^−1^
c3 mir	0.989	2.66 × 10^−1^
c2 cp biocarta	0.992	2.73 × 10^−1^
c5 GO bp	0.993	2.98 × 10^−1^
c4 cm	0.990	3.25 × 10^−1^
c4 cgn	0.992	3.41 × 10^−1^
c2 cp	0.993	3.77 × 10^−1^
c5 GO mf	0.992	3.89 × 10^−1^
c2 cgp	0.993	4.16 × 10^−1^
c7 immunologic signatures	0.977	4.50 × 10^−1^
c1 positional	0.991	4.58 × 10^−1^

**Table 2 ijms-22-11531-t002:** Cancer-related c2 reactome; 42 pathways and one sample z-test FDR q-value of pathways (pathways with FDR q-value < 1 × 10^−10^ are shown in this table; the FDR q-values of all 42 pathways are shown in Appendix A).

Pathway (c2 Reactome)	FDR (q-Value)
GPCR DOWNSTREAM SIGNALING	0
GPCR LIGAND BINDING	4.54 × 10^−100^
NEURONAL SYSTEM	9.86 × 10^−73^
SLC MEDIATED TRANSMEMBRANE TRANSPORT	1.11 × 10^−64^
SIGNALING BY EGFR IN CANCER	4.13 × 10^−55^
AXON GUIDANCE	6.81 × 10^−53^
PEPTIDE LIGAND BINDING RECEPTORS	1.31 × 10^−29^
DIABETES PATHWAYS	9.06 × 10^−20^
GASTRIN CREB SIGNALING PATHWAY VIA PKC AND MAPK	4.20 × 10^−17^
SIGNALING BY FGFR MUTANTS	9.46 × 10^−17^
COLLAGEN FORMATION	3.74 × 10^−16^
FATTY ACYL COA BIOSYNTHESIS	3.42 × 10^−15^
CIRCADIAN REPRESSION OF EXPRESSION BY REV ERBA	8.17 × 10^−12^
NFKB IS ACTIVATED AND SIGNALS SURVIVAL	1.73 × 10^−11^
ACETYLCHOLINE BINDING AND DOWNSTREAM EVENTS	6.24 × 10^−11^

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
