# Peer review of "Classification and Functional Analysis between Cancer and Normal Tissues Using Explainable Pathway Deep Learning through RNA-Sequencing Gene Expression"

_ijms, 2021, doi:10.3390/ijms222111531_

Round 1

Reviewer 1 Report

General comments

In this work, the authors present PathDeep, a deep-learning-based technique for clarification, and pathway enrichment. PathDeep aims to use the knowledge-based approach by incorporating pathway-gene mapping data in their enrichment analysis. Please see specific comments:

  1. a) major

1- The proposed approach, its novelty, and contribution compare to other existing tools are not clearly described.

2- There is no available code or demo for end-users to apply and test PathDeep.

3- Figure 6. Data processing and analysis flowchart can be extended to an overview figure to show a complete workflow of PathDeep.

4- On page 11, It is not clear what the authors try to explain “We created a PathDeep model with random biological functional relationships in 324 pathway layers and input genes to investigate how the biological functional relationship 325 affects the discrimination between cancerous and normal tissues (PathDeep random 326 linked).” I this used to calculate a p-value? 

5- in section 4.5 Selecting an Appropriate Pathway Database for PathDeep, it is not clear “PathDeep random linked 100 times” is used to calculate p-value using permutation test or not. 

  1. b) Minor

1- In figure 7, can a gene be a member of multiple pathways? 

2- An overview figure needs to include an abstract version of Figure 8 to understand the overall steps of PathDeep. Figure 8 is almost is the end of the manuscript until the readers can see what is happening in this s work. 

Author Response

Dear Reviewer 1, 
We have attached our response to your comments.

Please see the attachment file.

Sincerely, 

TaeJin Ahn

Reviewer 2 Report

Comments:

First of all, congratulate the authors for the excellent article. The article is very interesting and I would like to accept it in its present form.

Author Response

Dear Reviewer 2,

Thank you so much for your review.

Sincerely,

TaeJin Ahn

Reviewer 3 Report

In this work, the authors propose Pathway DNN (PathDeep) which uses the biological relationship between pathways and their component genes. The objective being to allow PathDeep to classify cancers or normal tissues based on gene expression data with favorable performance and to provide interpretation based on pathway knowledge.

The work seems interesting, nevertheless it presents important insufficiencies:

- Table 2 gives infinitesimally small values (of the order of E-100), how much credibility can we give them?

- What is the dispersion of all these values?

- On page 12, there are two unnumbered relations. Also, what does the symbol "*" mean?

- In the manuscript, there is no section reserved for conclusion and perspectives.

- Authors should enrich their introduction by drawing on the following text and citing references [1] and [2], they should also compare their work with reference [2]:

Cancer is one of the most feared and aggressive diseases in the world and is responsible for more than 9 million deaths universally. Deep learning has proven advantageous in solving cancer diagnostic or classification problems [1].

[1] Ouahabi, A.; Taleb-Ahmed, A. Deep learning for real-time semantic segmentation: Application in ultrasound imaging. Pattern Recognition Letters 2021144, 27–34.

Staging cancer early increases the chances of recovery. One staging technique is RNA sequence analysis [2]. 

[2] N. E. M. Khalifa, M. H. N. Taha, D. Ezzat Ali, A. Slowik and A. E. Hassanien, "Artificial Intelligence Technique for Gene Expression by Tumor RNA-Seq Data: A Novel Optimized Deep Learning Approach," in IEEE Access, vol. 8, pp. 22874-22883, 2020, doi: 10.1109/ACCESS.2020.2970210.

Author Response

Dear Reviewer 3,
We have attached our response to your comments.

Please see the attachment file.

Sincerely,

TaeJin Ahn

Round 2

Reviewer 1 Report

I appreciate the authors addressing my comments.

The software needs extensive work to be usable by others and requires documentation and tutorials. Hower it sounds the cancer analysis is the focus of this work and not the software. 
deepath explained as a general approach for pathway enrichment analysis using effect size in https://doi.org/10.3389/fimmu.2021.661437 can be referenced in this work since they do the same job in different ways and as they have similar names helps to clarify the differences. 

Author Response

Comments for Reviewer 1

The software needs extensive work to be usable by others and requires documentation and tutorials. However, it sounds the cancer analysis is the focus of this work and not the software. 

There are new updates in the documentation of PathDeep in the github. We added tutorial and code explanation for the user to understand how the PathDeep algorithm works.

Please find our updates by clicking the following URL.

https://github.com/sipark5340/PathDeep.

These following contents are updated in the github:

Description - PathDeep

How to Run

Input Files

Output Files

Run a Demo

Learning the PathDeep: PathDeep_example.py

Interpretation of PathDeep: Extract_PathDeep_gene_pathway_index.py

Troubleshooting

deepath explained as a general approach for pathway enrichment analysis using effect size in https://doi.org/10.3389/fimmu.2021.661437 can be referenced in this work since they do the same job in different ways and as they have similar names helps to clarify the differences. 

We reviewed deepath to follow your comment.

We found the mentioned paper that you have suggested which was about the research paper that utilized deepath. Researchers used the deepath to discover human gene and viral gene expression’s association with the disease (SLE: systemic lupus erythematosus). However, deepath methodology itself was not provided in the paper.

We tried to find technical details of deepath in other sources, but we were unable to find any except R code from the github. Thus, we are afraid that our understanding of deepath is very limited.

Based on the available information we have found, we think deepath is different to ours as it uses both of viral and human gene expression as an input. We also find deepath utilizes both linear and non-linear approaches to discover significantly associated pathways, but we couldn’t find what were the non-linear methods they exactly used to compare deepath to our deep learning approach.

As we were unable to have clear information about deepath, we think it is better not to mention their approaches in our manuscript to minimize potential ‘harmfulness’ of spreading unsure knowledge regarding deepath.

Reviewer 3 Report

The authors answered my questions and the manuscript was enhanced.

Author Response

Dear Reviewer 3, 

Thank you for your consideration.
According to your comments, we conducted moderate English changes. We also updated the github for users to use PathDeep algorithm.

Sincerely, 

Taejin Ahn